# Adaptive Depth Networks with Skippable Sub-Paths

**Woochul Kang**[*]
Department of Embedded Systems
Incheon National University
Yeonsu-gu, Incheon, South Korea, 22012
wchkang@inu.ac.kr

**Hyungseop Lee**
Department of Embedded Systems
Incheon National University
Yeonsu-gu, Incheon, South Korea, 22012
hhss0927@inu.ac.kr

## Abstract

Predictable adaptation of network depths can be an effective way to control inference latency and meet the resource condition of various devices. However, previous adaptive depth networks do not provide general principles and a formal explanation on why and which layers can be skipped, and, hence, their approaches are hard to be generalized and require long and complex training steps. In this paper, we present a practical approach to adaptive depth networks that is applicable to both convolutional neural networks (CNNs) and transformers with minimal training effort. In our approach, every hierarchical residual stage is divided into two sub-paths, and they are trained to acquire different properties through a simple self-distillation strategy. While the first sub-path is essential for hierarchical feature learning, the second one is trained to refine the learned features and minimize performance degradation even if it is skipped. Unlike prior adaptive networks, our approach does not train every target sub-network exhaustively. At test time, however, we can connect these sub-paths in a combinatorial manner to select sub-networks of various accuracy-efficiency trade-offs from a single network. We provide a formal rationale for why the proposed training method can reduce overall prediction errors while minimizing the impact of skipping sub-paths. We demonstrate the generality and effectiveness of our approach with both CNNs and transformers. Source codes are available at https://github.com/wchkang/depth

## 1 Introduction

Modern deep neural networks such as CNNs and transformers [1] provide state-of-the-art performance at high computational costs, and, hence, lots of efforts have been made to leverage those inference capabilities in various resource-constrained devices. Those efforts include compact architectures [2, 3], network pruning [4, 5], weight/activation quantization [6], knowledge distillation [7], to name a few. However, those approaches provide static accuracy-efficiency trade-offs, and, hence, it is infeasible to deploy one single model to meet devices with all kinds of resource-constraints.

There have been some attempts to provide predictable adaptability to neural networks by exploiting the redundancy in either network depths [8, 9], widths [10, 11], or both [12, 13]. However, one major difficulty with prior adaptive networks is that they are hard to train and require significantly longer training time than non-adaptive networks. For example, most adaptive networks select a fixed number of sub-networks of varying depths or width, and train them iteratively, mostly through self-distilling knowledge from the largest sub-network (also referred to as the *super-net*) [10, 11, 13, 14]. However, this exhaustive self-distillation takes long time and can generate conflicting training objectives for different parameter-sharing sub-networks, potentially resulting in worse performance [15, 16]. Moreover, unlike width-adaptation networks, there is no established principle for adapting depths since the impact of skipping individual layers has not been formally investigated.

---

[*]Corresponding author

38th Conference on Neural Information Processing Systems (NeurIPS 2024).

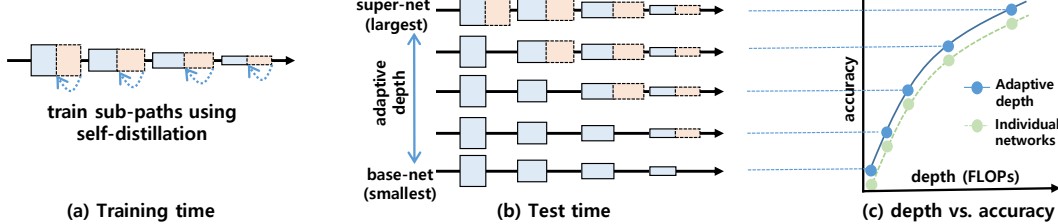

Figure 1: (a) During training, every residual stage of a network is divided into two sub-paths. The layers in every second (orange) sub-path are optimized to minimize performance degradation even if they are skipped. (b) At test time, these second sub-paths can be skipped in a combinatorial manner, allowing instant selection of various parameter sharing sub-networks. (c) The sub-networks selected from a single network form a better Pareto frontier than counterpart individual networks.

In this work, we introduce an architectural pattern and training method for adaptive depth networks that is generally applicable to various networks, e.g., CNNs and transformers. In the proposed adaptive depth networks, every residual stage is divided into two sub-paths and the sub-paths are trained to have different properties. While the first sub-paths are mandatory for hierarchical feature learning, the second sub-paths are optimized to incur minimal performance degradation even if they are skipped.

In order to enforce this property of the second sub-paths, we propose a simple self-distillation strategy, in which only the largest sub-network (or, *super-net*) and the smallest sub-network (or, *base-net*) are exclusively used as a teacher and a student, respectively, as shown in Figure-1-(a). The proposed self-distillation strategy does not require exhaustive training of every target sub-network, resulting in significantly shorter training time than prior adaptive networks. However, at test time, sub-networks with various depths can be selected instantly from a single network by connecting these sub-paths in a combinatorial manner, as shown in Figure 1-(b). Further, these sub-networks with varying depths outperform individually trained non-adaptive networks due to the regularization effect, as shown in Figure 1-(c).

In Section 3, we discuss the details of our architectural pattern and training algorithm, and show formally that the selected sub-paths trained with our self-distillation strategy are optimized to reduce prediction errors while minimally changing the level of input features. In Section 4, we empirically demonstrate that our adaptive depth networks outperform counterpart individual networks, both in CNNs and vision transformers, and achieve actual inference acceleration and energy-saving.

## 2   Related Work

**Adaptive Networks:** In most adaptive networks, parameter-sharing sub-networks are instantly selected by adjusting either widths, depths, or resolutions [8, 10, 11, 17, 13, 15, 18, 12, 19]. For example, slimmable neural networks adjust channel widths of CNN models on the fly for accuracy-efficiency trade-offs and they exploit switchable batch normalization to handle multiple sub-networks [10, 20, 11]. Transformer-based adaptive depth networks have been proposed for language models to dynamically skip some of the layers during inference [13, 9]. However, in these adaptive networks, every target sub-network with varying widths or depths need to be trained explicitly, incurring significant training overheads and potential conflicts between sub-networks.

**Dynamic Networks:** Dynamic networks [21] are another class of adaptive networks that exploit additional control networks or decision gates for input-dependent adaptation of CNN models [22, 23, 24, 25, 26] and transformers [27, 28, 29, 30]. In particular, most dynamic networks for depth-adaptation have some kinds of decision gates at every layer (or block) that determine if the layer can be skipped [31, 22, 32, 33]. These approaches are based on the thought that some layers can be skipped on 'easy' inputs. However, the learned policy for skipping layers is opaque to users and does not provide a formal description of which layers can be skipped for a given input. Therefore, the network depth cannot be controlled by users in a predictable manner to meet the resource condition of target devices.

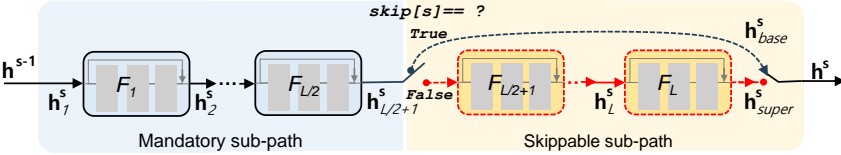

Figure 2: Illustration of a residual stage with two sub-paths. While the first (blue) sub-path is mandatory for hierarchical feature learning, the second (orange) sub-path can be skipped for efficiency. The layers in the skippable sub-path are trained to preserve the feature distribution from $\mathbf{h}_{base}^s$ to $\mathbf{h}_{super}^s$ using the proposed self-distillation strategy. Having similar distributions, either $\mathbf{h}_{base}^s$ or $\mathbf{h}_{super}^s$ can be provided as input $\mathbf{h}^s$ to the next residual stage. In the mandatory sub-path, another set of batch normalization operators, called *skip-aware BNs*, is exploited if the second sub-path is skipped. These sub-paths are building blocks to construct sub-networks of varying depths.

**Residual Blocks with Shortcuts:** Since the introduction of ResNets [34], residual blocks with identity shortcuts have received extensive attention because of their ability to train very deep networks, and have been chosen by many CNNs [35, 36] and transformers [1, 37, 38]. Veit et al. [39] argue that identity shortcuts make exponential paths and results in an ensemble of shallower sub-networks. This thought is supported by the fact that removing individual residual blocks at test time does not significantly affect performance, and it has been further exploited to train deep networks [40, 41]. Other works argue that identity shortcuts enable residual blocks to perform iterative feature refinement, where each block improves slightly but keeps the semantic of the representation of the previous layer [42, 43]. Our work build upon those views on residual blocks with shortcuts and further extend them for adaptive depth networks by introducing a training method that enforces the properties of residual blocks more explicitly for skippable sub-paths.

## 3  Adaptive Depth Networks

We first present the architecture and training details of adaptive depth networks. Then, we discuss the theoretic rationale for how depth adaptation can be achieved with minimal performance degradation.

### 3.1  Architectural Pattern for Depth Adaptation

In modern hierarchical networks, such as ResNets [34] and Swin transformers [38], there are typically 4 to 7 consecutive residual stages.[2] In a network with $N_r$ stages, the $s$-th ($s = 1, ..., N_r$) stage consists of $L$ (typically $L \in [3, 6]$) identical residual blocks that transform input features $\mathbf{h}_1^s$ additively to produce the output features $\mathbf{h}^s$, as follows:

$$\underbrace{\mathbf{h}_1^s}_{=\mathbf{h}^{s-1}} + \underbrace{F_1(\mathbf{h}_1^s) + ... + F_{L/2}^s(\mathbf{h}_{L/2}^s)}_{\mathbf{F}_{base}^s} + \underbrace{F_{L/2+1}(\mathbf{h}_{L/2+1}^s)... + F_L(\mathbf{h}_L^s)}_{\mathbf{F}_{skippable}^s} = \mathbf{h}^s \qquad (1)$$

While a block with a residual function $F_\ell$ ($\ell = 1, ..., L$) learns hierarchical features as traditional compositional networks [44], previous literature [43, 42] demonstrates that a residual function also tend to learn a function that refines already learned features at the same feature level. If a residual block mostly performs feature refinement while not changing the level of input features, the performance of the residual network is not significantly affected by dropping the block at test time [40, 41]. However, in typical residual networks, most residual blocks tend to refine features while learning new level features as well, and, hence, random dropping of residual blocks at test time degrades inference performance significantly. Therefore, we hypothesize that if some selected residual blocks can be encouraged explicitly during training to focus more on feature refinement, then these blocks can be skipped to save computation at marginal loss of prediction accuracy at test time.

To this end, we propose an architectural pattern for adaptive depth networks, in which every residual stage is divided into two sub-paths, or $\mathbf{F}_{base}^s$ and $\mathbf{F}_{skippable}^s$ as in Equation 1 and Figure 2. We train these two sub-paths to have different properties (Section 3.2). While $\mathbf{F}_{base}^s$ is trained to learn feature representation $\mathbf{h}_{base}^s$ ($= \mathbf{h}_{L/2+1}^s$) with no constraint, the second sub-path $\mathbf{F}_{skippable}^s$ is constrained

---

[2]For vision transformers, residual blocks/stages refer to encoder blocks/stages.

to preserve the feature level of $\mathbf{h}_{base}^s$ and only refine it to produce $\mathbf{h}_{super}^s (= \mathbf{h}_{L+1}^s)$. Since layers in $\mathbf{F}_{base}^s$ perform essential transformations for hierarchical feature learning, they cannot be bypassed during inference. In contrast, layers in $\mathbf{F}_{skippable}^s$ can be skipped for efficiency since they only refine $\mathbf{h}_{base}^s$. If $\mathbf{F}_{skippable}^s$ is skipped, then $\mathbf{h}_{base}^s$ becomes the input to the next residual stage. Therefore, $2^{N_r} (=\sum_{k=0}^{N_r} C(N_r, k))$ sub-networks with varying accuracy-efficiency trade-offs can be selected from a single network by choosing whether to skip $\mathbf{F}_{skippable}^s (s = 1, ..., N_r)$ or not (Table 6).

This architectural pattern is agnostic to the type of residual blocks. The residual function $F$ can be convolution layers for CNNs and self-attention + MLP layers for transformers. Table 5 in Appendix A.1 provides details on how this pattern is applied to both CNNs and transformers.

## 3.2 Training Sub-Paths with Self-Distillation

Preserving the feature level of $\mathbf{h}_{base}^s$ in $\mathbf{F}_{skippable}^s$ implies, more specifically, that two feature representations $\mathbf{h}_{base}^s$ and $\mathbf{h}_{super}^s$ have similar distributions over training input $\mathbf{X}$. Algorithm 1 shows our training method, in which $\mathbf{h}_{base}^s$ and $\mathbf{h}_{super}^s$ are encouraged to have similar distributions by including *Kullback-Leibler* (KL) divergence between them, or $D_{KL}(\mathbf{h}_{super}^s \| \mathbf{h}_{base}^s)$, in the loss function.

---

**Algorithm 1** Training algorithm for an adaptive depth network $\mathbf{M}$. The forward function of $\mathbf{M}$ accepts an extra argument, '*skip*', which controls the residual stages where their skippable sub-paths are skipped. For example, the smallest sub-network, or *base-net*, of $\mathbf{M}$ is selected by passing '*skip=[True, True, True, True]*' when the total number of residual stages, denoted by $N_r$, is 4.

---
1: Initialize an adaptive depth network $\mathbf{M}$
2: **for** $i = 1$ **to** $n_{iters}$ **do**
3:     Get next mini-batch of data $\mathbf{x}$ and label $\mathbf{y}$
4:     $optimizer.zero\_grad()$
5:     $\hat{\mathbf{y}}_{super}, \mathbf{h}_{super} = \mathbf{M}.forward(\mathbf{x}, \text{skip=[False, ..., False]})$     ▷ forward pass for *super-net*
6:     $loss_{super} = criterion(\mathbf{y}, \hat{\mathbf{y}}_{super})$
7:     $loss_{super}.backward()$
8:     $\hat{\mathbf{y}}_{base}, \mathbf{h}_{base} = \mathbf{M}.forward(\mathbf{x}, \text{skip=[True, ..., True]})$     ▷ forward pass for *base-net*
9:     $loss_{base} = [\sum_{s=1}^{N_r} D_{KL}(\mathbf{h}_{super}^s \| \mathbf{h}_{base}^s)]^* + D_{KL}(\hat{\mathbf{y}}_{super} \| \hat{\mathbf{y}}_{base})$     ▷ $[\ ]^*$ is optional
10:    $loss_{base}.backward()$     ▷ self-distillation of skippable sub-paths
11:    $optimizer.step()$
12: **end for**

---

In Algorithm 1, the largest and the smallest sub-networks of $\mathbf{M}$, which are called *super-net* and the *base-net*, respectively, are exploited. In steps 9-10, the hierarchical features (and outputs) from the super-net and the base-net are encouraged to have similar distributions through self-distillation. Self-distillation has been extensively used in prior adaptive networks [11, 10, 13]. However, their primary goal is to train every target sub-network. For example, on each iteration, sub-networks are randomly sampled from a large search space to act as either teachers or students [11], which takes significantly longer training time and can generate conflicting training objectives for different parameter-sharing sub-networks [15, 16]. In contrast, our method in Algorithm 1 focuses on training sub-paths by exclusively using the super-net as a teacher and the base-net as a student. By focusing on sub-paths, the training procedure in Algorithm 1 is significantly simplified and avoids potential conflicts among sub-networks. At test time, however, various sub-networks can be selected by connecting these sub-paths. The effect of this self-distillation strategy is investigated in Section 4.4.

Lastly, due to the architectural pattern of interleaving the mandatory and the skippable sub-paths, minimizing $D_{KL}(\hat{\mathbf{y}}_{super} \| \hat{\mathbf{y}}_{base})$ also minimizes $D_{KL}(\mathbf{h}_{super} \| \mathbf{h}_{base})$ implicitly. The ablation study in Section 4.4 shows that omitting the term $\sum_{s=1}^{N_r} D_{KL}(\mathbf{h}_{super}^s \| \mathbf{h}_{base}^s)$ in $loss_{base}$ has a minor effect on the performance. This implicit approach is useful when the extraction of intermediate features is tricky.

## 3.3 Analysis of Skippable Sub-Paths

**Formal Analysis**: $D_{KL}(\mathbf{h}_{super}^s \| \mathbf{h}_{base}^s)$ in the loss function $loss_{base}$ can be trivially minimized if residual blocks in $\mathbf{F}_{skippable}^s$ learn identity functions, or $\mathbf{h}_{base}^s + \mathbf{F}_{skippable}^s(\mathbf{h}_{base}^s) = \mathbf{h}_{base}^s$.

However, since the super-net is jointly trained with the loss function $loss_{super}$, the residual functions in $\mathbf{F}^s_{skippable}$ cannot simply be an identity function. Then, what do the residual functions in $\mathbf{F}^s_{skippable}$ learn during training? This can be further investigated through Taylor expansion [43]. For our adaptive depth networks, a loss function $\mathcal{L}$ used for training the super-net can be approximated with Taylor expansion as follows:

$$\mathcal{L}(\mathbf{h}^s_{super}) = \mathcal{L}\{\mathbf{h}^s_{base} + \mathbf{F}^s_{skippable}(\mathbf{h}^s_{base})\} \tag{2}$$

$$= \mathcal{L}\{\mathbf{h}^s_{base} + F_{L/2+1}(\mathbf{h}^s_{L/2+1}) + ... + F_{L-1}(\mathbf{h}^s_{L-1}) + F_L(\mathbf{h}^s_L)\} \tag{3}$$

$$\approx \mathcal{L}\{\mathbf{h}^s_{base} + F_{L/2+1}(\mathbf{h}^s_{L/2+1}) + ... + F_{L-1}(\mathbf{h}^s_{L-1})\} + F_L(\mathbf{h}^s_L) \cdot \frac{\partial \mathcal{L}(\mathbf{h}^s_L)}{\partial \mathbf{h}^s_L} + \mathcal{O}(F_L(\mathbf{h}^s_L)) \tag{4}$$

In Equation 4, the loss function is expanded around $\mathbf{h}^s_L$, or $\mathbf{h}^s_{base} + ... + F_{L-1}(\mathbf{h}^s_{L-1})$. Only the first order term is left and all high order terms, such as $F_L(\mathbf{h}^s_L)^2 \cdot \frac{\partial^2 \mathcal{L}(\mathbf{h}^s_L)}{2\partial(\mathbf{h}^s_L)^2}$, are absorbed in $\mathcal{O}(F_L(\mathbf{h}^s_L))$.

The high-order terms in $\mathcal{O}(F_L(\mathbf{h}^s_L))$ can be ignored if $F_L(\mathbf{h}^s_L)$ has a small magnitude. In typical residual networks, however, every layer is trained to learn new features with no constraint, and, hence, there is no guarantee that $F_L(\mathbf{h}^s_L)$ have small magnitude. In contrast, in our adaptive depth networks, the residuals in $\mathbf{F}^s_{skippable}$ are explicitly enforced to have small magnitude through the proposed self-distillation strategy (refer to Figure 3 for empirical evidence). As a result, the terms in $\mathcal{O}(F_L(\mathbf{h}^s_L))$ can be ignored for the approximation. If we similarly keep expanding the loss function around $\mathbf{h}^s_j$ ($j = L/2 + 1, ..., L$) while ignoring high order terms, we obtain the following approximation:

$$\mathcal{L}(\mathbf{h}^s_{super}) \approx \mathcal{L}(\mathbf{h}^s_{base}) + \sum_{j=L/2+1}^{L} F_j(\mathbf{h}^s_j) \cdot \frac{\partial \mathcal{L}(\mathbf{h}^s_j)}{\partial \mathbf{h}^s_j} \tag{5}$$

In Equation 5, minimizing the loss $\mathcal{L}(\mathbf{h}^s_{super})$ during training drives $F_j(\mathbf{h}^s_j)$ ($j = L/2 + 1, ..., L$) in the negative half space of $\frac{\partial \mathcal{L}(\mathbf{h}^s_j)}{\partial \mathbf{h}^s_j}$ to minimize the dot product between $F_j(\mathbf{h}^s_j)$ and $\frac{\partial \mathcal{L}(\mathbf{h}^s_j)}{\partial \mathbf{h}^s_j}$. This implies that every residual function in $\mathbf{F}^s_{skippable}$ is optimized to learn a function that has a similar effect to gradient descent:

$$F_j(\mathbf{h}^s_j) \simeq - \frac{\partial \mathcal{L}(\mathbf{h}^s_j)}{\partial \mathbf{h}^s_j} \ (j = L/2 + 1, ..., L) \tag{6}$$

In other words, the residual functions in the skippable sub-paths reduce the loss $\mathcal{L}(\mathbf{h}^s_{base})$ iteratively during inference while preserving the feature distribution of $\mathbf{h}^s_{base}$.

Considering this result, we can conjecture that, with our architectural pattern and self-distillation strategy, layers in $\mathbf{F}^s_{skippable}$ learn functions that refine input features $\mathbf{h}^s_{base}$ iteratively for better inference accuracy while minimally changing the distribution of $\mathbf{h}^s_{base}$.

**Empirical Analysis**: We can estimate how much the distribution of input $\mathbf{h}$ is transformed by residual function $F$ by measuring $||F(\mathbf{h})||_2/||\mathbf{h}||_2$ at each residual block. Figure 3 illustrates $||F(\mathbf{h})||_2/||\mathbf{h}||_2$ at every residual block in the baseline ResNet50 and our ResNet50-ADN that is trained according to Algorithm 1.[3] Our ResNet50-ADN exhibits greater transformation than ResNet50 in the mandatory sub-paths (blue areas) and less transformation in the skippable sub-paths (orange areas). This result demonstrates that our self-distillation strategy in Algorithm 1 effectively trains the skippable sub-paths to minimally change the input distribution. As a result, the blocks in skippable sub-paths can be skipped with less impact on performance.

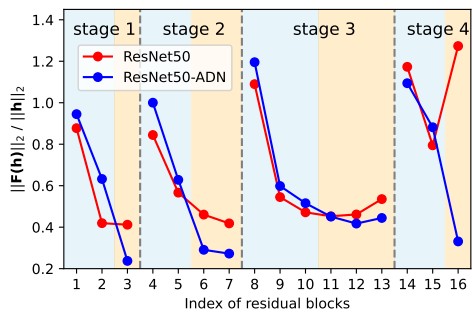

Figure 3: $||F(\mathbf{h})||_2/||\mathbf{h}||_2$ at residual blocks. In ours, skippable sub-paths (orange areas) minimally change the distribution of input $\mathbf{h}$.

---

[3]ImageNet validation dataset is used for this experiment.

### 3.4 Skip-Aware Batch/Layer Normalization

Originally, batch normalization (BN) [45] was proposed to handle internal covariate shift during training non-adaptive networks by normalizing features. In our adaptive depth networks, however, internal covariate shifts can occur during inference in mandatory sub-paths if different sub-networks are selected. To handle potential internal covariate shifts, switchable BN operators, called *skip-aware BNs*, are used in mandatory sub-paths. For example, at each residual stage, two sets of BNs are available for the mandatory sub-path, and they are switched depending on whether its skippable sub-path is skipped or not.

The effectiveness of switchable BNs has been demonstrated in networks with adaptive widths [46, 11] and adaptive resolutions [47]. However, in previous adaptive networks, $N$ sets of switchable BNs are required in every layer to support $N$ parameter-sharing sub-networks. Such a large number of switchable BNs not only requires more parameters, but also makes the training process complicated since $N$ sets of switchable BNs need to be trained iteratively during training. In contrast, in our adaptive depth networks, every mandatory sub-path needs only two sets of switchable BNs, regardless of the number of supported sub-networks. This reduced number of switchable BNs significantly simplifies the training process as shown in Algorithm 1. Furthermore, the amount of parameters for skip-aware BNs is negligible. For instance, in ResNet50, skip-aware BNs increase the parameters by 0.07%.

Transformers [1, 38] exploit layer normalization (LN) instead of BNs and naive replacement of LNs to BNs incurs instability during training [48]. Therefore, for our adaptive depth transformers, we apply switchable LN operators in mandatory sub-paths instead of switchable BNs.

## 4 Experiments

We use networks both from CNNs and vision transformers as base models to apply the proposed architecture pattern: MobileNet V2 [35] is a lightweight CNN model, ResNet [34] is a larger CNN model, and ViT [37] and Swin-T [38] are representative vision transformers. All base models except ViT have hierarchical stages, each with $2 \sim 6$ residual blocks. So, according to the proposed architectural pattern, every stage is evenly divided into 2 sub-paths for depth adaptation. ViT does not define hierarchical stages and all residual encoder blocks have same spatial dimensions. Therefore, we divide 12 encoder blocks into 4 groups, resembling other residual networks, and select the last encoder block of each group as a skippable sub-path. Details are in Table 5 in Appendix A.1. These models are trained according to Algorithm 1. For self-distillation, only the final outputs from the super-net and the base-net, or $\hat{\mathbf{y}}_{super}$ and $\hat{\mathbf{y}}_{base}$ respectively, are used since exploiting intermediate features for explicit self-distillation has a marginal impact on performance (Section 4.4).

We use the suffix *'-ADN'* to denote our adaptive depth networks. A series of boolean values in parentheses denotes a specific sub-network used for evaluation; each boolean value represents the residual stage where its skippable sub-path is skipped. For example, ResNet50-ADN (FFFF) and ResNet50-ADN (TTTT) correspond to the super-net and the base-net of ResNet50-ADN, respectively.

### 4.1 ImageNet Classification

We evaluate our method on ILSVRC2012 dataset [49] that has 1000 classes. The dataset consists of 1.28M training and 50K validation images. For CNN models, we follow most training settings in the original papers [34, 35], except that ResNet models are trained for 150 epochs. ViT and Swin-T are trained for 300 epochs, following DeiT's training recipe [50, 51]. For Swin-T-ADN, we disable stochastic depths [40] for the mandatory sub-paths since the strategy of random dropping of residual blocks conflicts with our approach to skipping sub-paths. For fair comparison, our adaptive depth networks and corresponding individual networks are trained in the same training settings.

The results in Figure 4-(a) show that our adaptive depth networks outperform counterpart individual networks even though many sub-networks share parameters in a single model. Further, our results with vision transformers demonstrate that our approach is generally applicable and compatible with their state-of-the-art training techniques such as DeiT's training recipe [50]. We conjecture that this performance improvement results from effective distillation of knowledge from $\mathbf{h}_{super}^{s}$ to $\mathbf{h}_{base}^{s}$ at each residual stage and the iterative feature refinement at skippable sub-paths, shown in Equation 6.

| Model | Params (M) | FLOPs (G) | Acc (%) |
|---|---|---|---|
| **ResNet50-ADN (FFFF)** | 25.58 | 4.11 | 77.6 |
| **ResNet50-ADN (TTTT)** | | 2.58 | 76.1 |
| ResNet50 | 25.56 | 4.11 | 76.7 |
| ResNet50-Base | 17.11 | 2.58 | 75.0 |
| **MbV2-ADN (FFFFF)** | 3.53 | 0.32 | 72.5 |
| **MbV2-ADN (TTTTT)** | | 0.22 | 70.6 |
| MbV2 | 3.50 | 0.32 | 72.1 |
| MbV2-Base | 2.98 | 0.22 | 70.2 |
| **ViT-b/16-ADN (FFFF)** | 86.59 | 17.58 | 81.4 |
| **ViT-b/16-ADN (TTTT)** | | 11.76 | 80.6 |
| ViT-b/16 | 86.57 | 17.58 | 81.1 |
| ViT-b/16-Base | 67.70 | 11.76 | 78.7 |
| **Swin-T-ADN (FFFF)** | 28.30 | 4.49 | 81.6 |
| **Swin-T-ADN (TTTT)** | | 2.34 | 78.0 |
| Swin-T | 28.29 | 4.49 | 81.5 |
| Swin-T-Base | 15.34 | 2.34 | 77.4 |

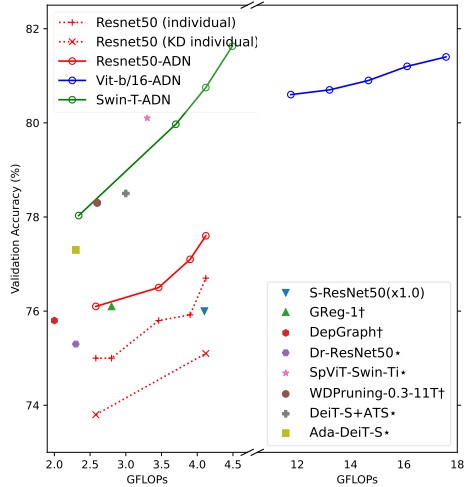

(a) Results on ImageNet  (b) Pareto frontiers of our networks

Figure 4: (a) Results on ImageNet validation dataset. Networks with the suffix '-Base' have the same depths as the base-nets of corresponding adaptive depth networks. (b) Pareto frontiers formed by the sub-networks of our adaptive depth networks. ResNet50 (individual) and ResNet50 (KD individual) are non-adaptive networks having same depths as the sub-networks of ResNet50-ADN.

Table 1: Our base-nets are compared with state-of-the-art efficient inference methods. † denotes static pruning methods, ∗ denotes width-adaptation networks, and ⋆ denotes input-dependent dynamic networks. While these approaches exploit various non-canonical training techniques, such as iterative retraining, our base-nets are instantly selected from adaptive depth networks without fine-tuning.

| Model | FLOPs | ↓FLOPs | Acc@1 |
|---|---|---|---|
| GReg-1 [55]† | 2.8G | 33% | 76.1% |
| DepGraph [56]† | 2.0G | 51% | 75.8% |
| DR-ResNet50 ($\alpha$=2.0) [47]⋆ | 2.3G | 44% | 75.3% |
| **ResNet50-ADN (TTTT)** | 2.6G | 37% | 76.1% |
| AlphaNet-0.75x [11] ∗ | 0.21G | 34% | 70.5% |
| **MbV2-ADN (TTTTT)** | 0.22G | 32% | 70.6% |
| WDPruning-0.3-11 [57]† | 2.6G | - | 78.3% |
| X-Pruner [58] † | 3.2G | 28% | 80.7% |
| Ada-DeiT-S [27]⋆ | 2.3G | - | 77.3% |
| SPViT-Swin-Ti [59]⋆ | 3.3G | 27% | 80.1% |
| **Swin-T-ADN (TTTT)** | 2.3G | 48% | 78.0% |

Figure 4-(b) shows Pareto frontiers formed by selected sub-networks of our adaptive depth networks; Table 6 in Appendix A.2 shows the performance of all sub-networks. In Figure 4-(b) and Table 1, several state-of-the-art efficient inference methods and dynamic networks are compared with our base-networks. The result demonstrates that our adaptive depth networks show comparable performance across a range of FLOPs. In Figure 4-(b), it should be noted that individual ResNets trained with knowledge distillation has worse performance than individual ResNets. As reported in previous works, successful knowledge distillation requires a patient and long training [52], and straightforward knowledge distillation using ImageNet does not improve the performance of student models [53, 54]. In contrast, our ResNet50-ADN trained with the proposed self-distillation strategy achieves better performance than counterpart ResNets. This demonstrates that the high performance of adaptive depth networks does not simply come from distillation effect.

## 4.2 Training Cost

One of the main advantages of our adaptive depth networks is the significantly lower training effort needed compared to previous adaptive networks. In Algorithm 1, only one additional forward and backward pass of the smallest sub-network, or the base-net, is required for self-distillation at each

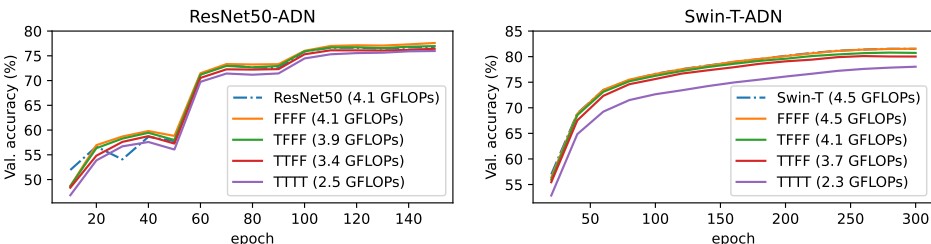

Figure 5: Validation accuracy of sub-networks of our adaptive depth networks during training. Many sub-networks of varying depths become available from a single network even though most of them are not explicitly trained.

Table 2: Training time (1 epoch), measured on Nvidia RTX 4090 (batch size: 128). AlphaNet* is configured to have similar FLOPs to MbV2 and only adjusts its depth to select sub-networks.

| Model | 1 epoch (min) | # of sub-nets |
|---|---|---|
| ResNet50 | 32.9 | - |
| ResNet50-Base | 22.7 | - |
| **ResNet50-ADN (ours)** | 54.7 | $2^4$ depths |
| MSDNet [8] | 67.1 | 9 depths |
| S-ResNet50 [10] | 82.5 | 4 widths |
| MbV2 | 13.1 | - |
| MbV2-Base | 10.0 | - |
| **MbV2-ADN (ours)** | 22.5 | $2^5$ depths |
| AlphaNet* [11] | 230.5 | 216 depths |

training iteration. For most CNNs and transformer networks, the original training settings, such as training schedules and hyperparameters, can be used with minimal changes. However, as shown in Figure 5, many sub-networks of varying depths become available from a single network by connecting sub-paths in a combinatorial manner even though they are not trained explicitly.

Table 2 shows that, at every training epoch, our adaptive depth networks require a comparable amount of time as training two separate networks combined. This is because our training method trains sub-paths, rather than sub-networks, by exploiting only the super-net and the base-net. In contrast, the compared adaptive networks require much longer training time than ours since they have to explicitly apply self-distillation to all target sub-networks. For example, on every training iteration, AlphaNet [11] randomly samples sub-networks from its search space for self-distillation. Although the prior works we compared may seem outdated, they remain relevant and representative because there has been little progress in improving the training cost of adaptive networks.

While AlphaNet [11] supports a significantly larger number of sub-networks, our primary goal is not to maximize the number of sub-networks. Instead, our objective is to provide better performance Pareto with a few useful sub-networks, as illustrated in Figure 4-(b). Since the classification performance of sub-networks does not always scale proportionally with their FLOPS, most sub-networks end up being ineffective. For example, although ResNet50-ADN has $2^4$ sub-networks in its search space, Table 6 in Appendix A.2 shows that some shallower sub-networks outperform deeper sub-networks.

### 4.3 On-Device Performance

While the Pareto frontiers formed by sub-networks demonstrate theoretic performance, inference acceleration in actual devices is more important in practice for effective control of inference latency and energy consumption.

Figure 6-(a) shows the performance on Nvidia Jetson Orin Nano. The inference latency and energy consumption of ResNet50-ADN is compared to S-ResNet50, a representative width-adaptation network. The result shows that depth-adaptation of ResNet50-ADN is highly effective in accelerating inference speeds and reducing energy consumption. Although our ResNet50-ADN has a limited FLOPs adaptation range, reducing FLOPs by 38% through depth adaptation reduces both inference latency and energy consumption by 35%. In contrast, even though S-ResNet50 can reduce FLOPs by up to 93% by adjusting its width, it only achieves up to 9% acceleration in practice.

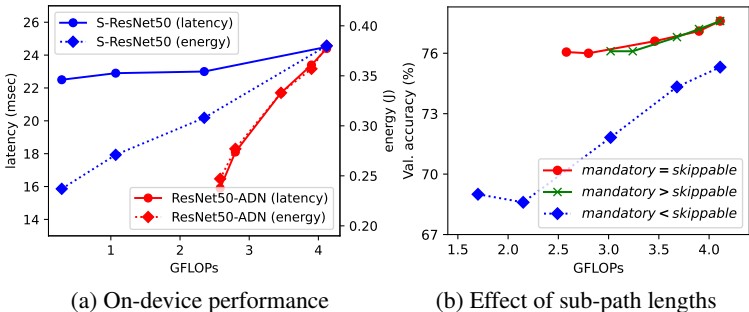

|     (a) On-device performance     |     (b) Effect of sub-path lengths     |

Figure 6: (a) Inference latency and energy consumption of adaptive networks, measured on Nvidia Jetson Orin Nano (batch size: 1) (b) Pareto frontiers of three ResNet50-ADNs, each trained with varying ratios between mandatory and skippable sub-paths. Total number of blocks remains unchanged.

## 4.4 Ablation Study

Table 3: Ablation analysis with ResNet50-ADN and ViT-b/32-ADN. Applied components are checked. ↓ and ↑ in parentheses are comparisons to non-adaptive individual networks. By default, only the outputs, or $\hat{\mathbf{y}}_{super}$ and $\hat{\mathbf{y}}_{base}$, are used for self-distillation. The last row with double check marks shows the results when both intermediate features and outputs are used for self-distillation.

| self-distllation. | skip-aware BNs/LNs | ResNet50-ADN Acc@1 (%) | | ViT-b/32-ADN Acc@1 (%) | |
|---|---|---|---|---|---|
| | | FFFF | TTTT | FFFF | TTTT |
| | | 75.2% (↓ 1.5%) | 72.2% (↓ 2.8%) | 75.7% (↓0.2%) | 74.1% (↑ 0.3%) |
| ✓ | | 76.1% (↓ 0.6%) | 74.9 % (↓ 0.1%) | 76.4% (↑0.5%) | 74.3% (↑0.5%) |
| | ✓ | 76.6% (↓ 0.1%) | 75.1% (↑ 0.1%) | 76.0% (↑0.1%) | 74.3% (↑0.5%) |
| ✓ | ✓ | 77.6% (↑ 0.9%) | 76.1% (↑ 1.1%) | 76.6% (↑ 0.8%) | 74.3% (↑0.5%) |
| ✓✓ | ✓ | 77.3% (↑ 0.6%) | 76.2% (↑ 1.2%) | | |

We first investigate the influence of two key components of the proposed adaptive depth networks: (1) self-distillation of sub-paths and (2) skip-aware BNs/LNs. When our self-distillation method is not applied, the loss of the base-net, or $loss_{base}$, in Algorithm 1 is modified to $criterion(\mathbf{y}, \hat{\mathbf{y}}_{base})$. Table 3 shows the results. For ResNet50-ADN, when neither of them is applied, the inference accuracy of the super-net and the base-net is significantly lower than non-adaptive individual networks by 1.5% and 2.8%, respectively. This result shows the difficulty of joint training sub-networks for adaptive networks. When one of the two components is applied individually, the performance is still slightly worse than individual networks'. When both self-distillation and skip-aware BNs are applied together, ResNet50-ADN achieves significantly better performance than individual networks, both in the super-net and the base-net. The last row, with double check marks, shows that exploiting intermediate features as well as softmax outputs for self-distillation has minor impact on performance.

Table 4: Comparison of self-distillation strategies. Our approach (in bold) uses exclusively the super-net and the base-net as a teacher and a student, respectively.

| sub-nets | | ResNet50-ADN Acc@1 (%) | | | | | ViT-b/32-ADN Acc@1 (%) | | | | |
|---|---|---|---|---|---|---|---|---|---|---|---|
| Teacher | Student | FFFF | TFFF | TTFF | TTTF | TTTT | FFFF | TFFF | TTFF | TTTF | TTTT |
| **FFFF** | **TTTT** | **77.6** | **77.1** | **76.5** | **76.0** | **76.1** | **76.6** | **76.0** | **75.5** | **74.5** | **74.3** |
| FFFF | Random | 77.1 | 76.7 | 76.4 | 75.5 | 74.8 | 75.2 | 74.7 | 73.9 | 72.9 | 71.1 |
| Random | TTTT | 75.5 | 75.5 | 75.4 | 75.0 | 74.9 | 72.0 | 72.0 | 71.9 | 71.8 | 71.7 |
| Random | Random | 75.4 | 75.2 | 75.2 | 74.9 | 74.6 | 70.8 | 70.7 | 70.6 | 70.4 | 70.3 |

**Self-Distillation Strategies**: Our self-distillation approach in Algorithm 1 exploits only two sub-networks exclusively as a teacher and a student. Specifically, the super-nets, or FFFF, acts as the the teacher and the base-nets, or TTTT, becomes the student. The purpose of this strategy is not to train only those two sub-networks, but rather to train skippable sub-paths in a way that minimally modifies

the feature distribution, as demonstrated in Figure 3. To investigate the effect of self-distillation strategies, we conduct an experiment in Table 4. In every training iteration, rather than exclusively using FFFF and TTTT sub-networks for self-distillation, we randomly sample either a teacher, a student, or both from $2^4$ sub-networks. These randomly sampled sub-networks are trained explicitly through self-distillation. However, as shown in Table 4, all sub-networks, both from ResNet50-ADN and ViT-b/32-ADN, perform significantly better when our self-distillation strategy is applied for training. Even though most of our sub-networks (such as TFFF, TTFF, and TTTF) are instantly constructed at test time without explicit training, they still outperform their counterpart sub-networks trained explicitly through random sampling. This result demonstrates that our method of training sub-paths is more effective than training target sub-networks themselves.

**Lengths of Mandatory Sub-Paths**: In Figure 6-(b), we investigate the impact of varying the ratio of lengths between the mandatory and the skippable sub-paths. (Details are in Table 7.) If mandatory sub-paths become shorter than skippable sub-paths, shallower sub-networks can be selected since more layers can be skipped. However, the result in Figure 6-(b) shows that this configuration (shown in blue line) significantly degrades the performance of all sub-networks. Since every sub-network shares parameters of mandatory sub-paths, low inference capability of shallow mandatory sub-paths affects all sub-networks. This implies that maintaining certain depths in mandatory sub-paths is crucial for effective inference. Conversely, further increasing the length of mandatory sub-paths (shown in green line) does not further improve performance and only reduces the range of depth adaptation.

## 5   Conclusions

We propose a practical approach to adaptive depth networks that can be applied to both CNNs and transformers with minimal training effort. We provide a general principle and a formal explanation on how depth adaptation can be achieved with minimal performance degradation. Under this principle, our approach can avoid typical exhaustive training of target sub-networks and instead focus on optimizing the sub-paths of the network to have specific properties. At test time, these sub-paths can be connected in a combinatorial manner to construct sub-networks with various accuracy-efficiency trade-offs from a single network. Experimental results show that these sub-networks form a better Pareto frontier than non-adaptive baseline networks and achieve actual inference acceleration. We anticipate that these advances will facilitate the practical application of adaptive networks.

## Acknowledgments and Disclosure of Funding

We thank the anonymous reviewers for their constructive comments and suggestions. This work was supported by the National Research Foundation of Korea (NRF) Grants Funded by the Ministry of Science and ICT under Grant NRF-2022R1F1A1074211.

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

# A Appendix: Detailed Settings and Evaluation Results

## A.1 Detailed Architectures

| | number of mandatory blocks | number of skippable blocks | total blocks |
|---|---|---|---|
| stage 1 | 2 | 1 | 3 |
| stage 2 | 2 | 2 | 4 |
| stage 3 | 3 | 3 | 6 |
| stage 4 | 2 | 1 | 3 |

(a) ResNet50-ADN

| | number of mandatory blocks | number of skippable blocks | total blocks |
|---|---|---|---|
| stage 1 | 2 | 1 | 3 |
| stage 2 | 2 | 1 | 3 |
| stage 3 | 2 | 1 | 3 |
| stage 4 | 2 | 1 | 3 |

(b) Vit-b/16-ADN

| | number of mandatory blocks | number of skippable blocks | total blocks |
|---|---|---|---|
| stage 1 | 1 | - | 1 |
| stage 2 | 1 | 1 | 2 |
| stage 3 | 2 | 1 | 3 |
| stage 4 | 2 | 2 | 4 |
| stage 5 | 2 | 1 | 3 |
| stage 6 | 2 | 1 | 3 |
| stage 7 | 1 | - | 1 |

(c) MbV2-ADN

| | number of mandatory blocks | number of skippable blocks | total blocks |
|---|---|---|---|
| stage 1 | 1 | 1 | 2 |
| stage 2 | 1 | 1 | 2 |
| stage 3 | 3 | 3 | 6 |
| stage 4 | 1 | 1 | 2 |

(d) Swin-T-ADN

Table 5: Each stage of base models is evenly divided into two sub-paths; the first is mandatory and the other is skippable. Since ViT does not define hierarchical stages, 12 identical encoder blocks are divided into 4 stages, resembling other residual networks for vision tasks.

## A.2 Performance of Sub-Networks

| sub-network | FLOPs (G) | Acc@1 (%) |
|---|---|---|
| FFFF | 4.11 | **77.6** |
| **T**FFF | 3.90 | **77.1** |
| F**T**FF | 3.68 | 76.5 |
| FF**T**F | 3.46 | 75.6 |
| FFF**T** | 3.90 | 76.7 |
| **TT**FF | 3.46 | **76.5** |
| **T**F**T**F | 3.24 | 75.3 |
| **T**FF**T** | 3.68 | 76.4 |
| F**TT**F | 3.02 | 75.8 |
| F**T**F**T** | 3.46 | 75.9 |
| FF**TT** | 3.25 | 75.6 |
| **TTT**F | 2.80 | 75.9 |
| **TT**F**T** | 3.24 | 75.8 |
| **T**F**TT** | 3.02 | 75.3 |
| F**TTT** | 2.80 | **76.0** |
| **TTTT** | 2.58 | **76.1** |

(a) ResNet50-ADN

| sub-network | FLOPs (G) | Acc@1 (%) |
|---|---|---|
| FFFF | 17.58 | **81.4** |
| **T**FFF | 16.20 | 81.1 |
| F**T**FF | 16.20 | 81.0 |
| FF**T**F | 16.20 | 80.6 |
| FFF**T** | 16.20 | **81.2** |
| **TT**FF | 14.67 | **80.9** |
| **T**F**T**F | 14.67 | 80.4 |
| **T**FF**T** | 14.67 | 80.9 |
| F**TT**F | 14.67 | 80.5 |
| F**T**F**T** | 14.67 | 80.9 |
| FF**TT** | 14.67 | 80.6 |
| **TTT**F | 13.21 | 80.5 |
| **TT**F**T** | 13.21 | **80.7** |
| **T**F**TT** | 13.21 | 80.5 |
| F**TTT** | 13.21 | 80.6 |
| **TTTT** | 11.76 | **80.6** |

(b) ViT-b/16-ADN

Table 6: FLOPs and ImageNet validation accuracy of sub-networks. Only super-net (or, FFFF) and base-net (or, TTTT) are trained explicitly. Sub-networks in the middle can be selected at test time without explicit training. The highest accuracy in each group is shown in bold.

### A.3 Varying the Ratio of Sub-Path Lengths in ResNet50-ADN

| | number of mandatory blocks | number of skippable blocks | total blocks |
|---|---|---|---|
| stage 1 | 1 | 2 | 3 |
| stage 2 | 1 | 3 | 4 |
| stage 3 | 2 | 4 | 6 |
| stage 4 | 1 | 2 | 3 |

(a) # of mandatory < # of skippable

| | number of mandatory blocks | number of skippable blocks | total blocks |
|---|---|---|---|
| stage 1 | 2 | 1 | 3 |
| stage 2 | 3 | 1 | 4 |
| stage 3 | 4 | 2 | 6 |
| stage 4 | 2 | 1 | 3 |

(b) # of mandatory > # of skippable

Table 7: The configurations of ResNet50-ADNs with different proportions between mandatory and skippable sub-paths. Total number of blocks at each stage remains unchanged.

### A.4 Varying the Ratio of Sub-Path Lengths in Vit-b/16-ADN

Each stage of Vit-b/16-ADN has 3 encoder blocks and, by default, we select every last encoder block as a skippable sub-path. Therefore every stage has two mandatory blocks and 1 skipable blocks as shown in Table 5-(b). Figure 7-(a) shows different configuration where every last two blocks of the stages become skippable. With this configuration in Figure 7-(a), we can select much smaller sub-networks. For example, the smallest sub-network, or the base-net, of Vit-b/15-ADN has only 4 mandatory blocks and it requires 5.82 GFLOPs. However, the result in Figure 7-(b) shows that this configuration significantly degrades performance of all sub-networks. As demonstrated with ResNet50-ADN in Figure 6-(b), the low inference capability of shallow mandatory sub-paths affects all sub-networks. This result again shows that maintaining certain depths in mandatory sub-paths is crucial for effective inference.

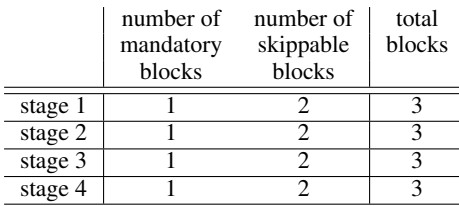

| | number of mandatory blocks | number of skippable blocks | total blocks |
|---|---|---|---|
| stage 1 | 1 | 2 | 3 |
| stage 2 | 1 | 2 | 3 |
| stage 3 | 1 | 2 | 3 |
| stage 4 | 1 | 2 | 3 |

(a) # of mandatory < # of skippable

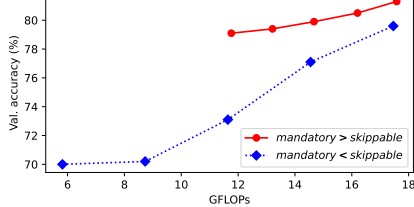

(b) Pareto frontiers of Vit-b/16-ADN

Figure 7: (a) The configuration of Vit-b/16-ADN with longer skippable sub-paths. (b) Pareto-frontier when different length ratios between the mandatory and the skippable sub-paths are applied.

## B Appendix: More Experiments and Analysis

### B.1 Object Detection and Instance Segmentation

Table 8: Object detection and instance segmentation results on MS COCO dataset.

| Detector | Backbone | FLOPs | Individual Networks | | **ResNet50-ADN (ours)** | |
|---|---|---|---|---|---|---|
| | | | Box AP | Mask AP | **Box AP** | **Mask AP** |
| Faster-RCNN | ResNet50 | 207.07G | 36.4 | | 37.8 | |
| [60] | ResNet50-Base | 175.66G | 32.4 | | 34.0 | |
| Mask-RCNN | ResNet50 | 260.14G | 37.2 | 34.1 | 38.3 | 34.1 |
| [61] | ResNet50-Base | 228.73G | 32.7 | 29.9 | 34.1 | 31.2 |
| RetinaNet | ResNet50 | 151.54G | 36.4 | | 37.4 | |
| [62] | ResNet50-Base | 132.04G | 31.7 | | 35.2 | |

In order to investigate the generalization ability of our approach, we use MS COCO 2017 datasets on object detection and instance segmentation tasks using representative detectors. We compare

individual ResNet50 and our adaptive depth ResNet50-ADN as backbone networks of the detectors. For training of detectors, we use Algorithm 1 with slight adaptation. For object detection, the intermediate features $\mathbf{h}_{base}^s$ and $\mathbf{h}_{super}^s (s = 1..N_r)$ can be obtained directly from backbone network's feature pyramid networks (FPN) [63], and, hence, a wrapper function is not required to extract intermediate features. All networks are trained on `train2017` for 12 epochs from ImageNet pretrained weights, following the training settings suggested in [63]. Table 8 shows the results on `val2017` containing 5000 images. Our adaptive depth backbone networks still outperform individual static backbone networks in terms of COCO's standard metric AP.

## B.2 Visual Analysis of Sub-Paths

To investigate how our training method affects feature representations in the mandatory and the skippable sub-paths, we visualize the activation of 3rd residual stage of ResNet50-ADN using Grad-CAM [64]. The 3rd residual stage of ResNet50-ADN has 6 residual blocks and the last three blocks are skippable. In Figure 8-(a), the activation regions of original ResNet50 changes gradually across all consecutive blocks. In contrast, in Figure 8-(b), ResNet50-ADN(FFFF), or super-net, manifests very different activation regions in two sub-paths. In the first three residual blocks, we can observe lots of hot activation regions in wide areas, suggesting active learning of new level features. In contrast, significantly less activation regions are found in the skippable last three blocks and they are gradually concentrated around the target object, demonstrating the refinement of learned features. While ResNet50-ADN(TTTT), or base-net, shares the parameters with the super-net in the first 3 mandatory blocks, their activation regions are very different from the super-net's. This is because while the super-net and the base-net share parameters in the non-skippable mandatory blocks, they use different batch normalization operators in the mandatory sub-paths. Further, in Figure 8-(c), we can observe that the final activation map of the base-net is very similar to the super-net's final activation map in Figure 8-(b). This implies that they have similar distributions for the same inputs, as suggested in Section 3.3.

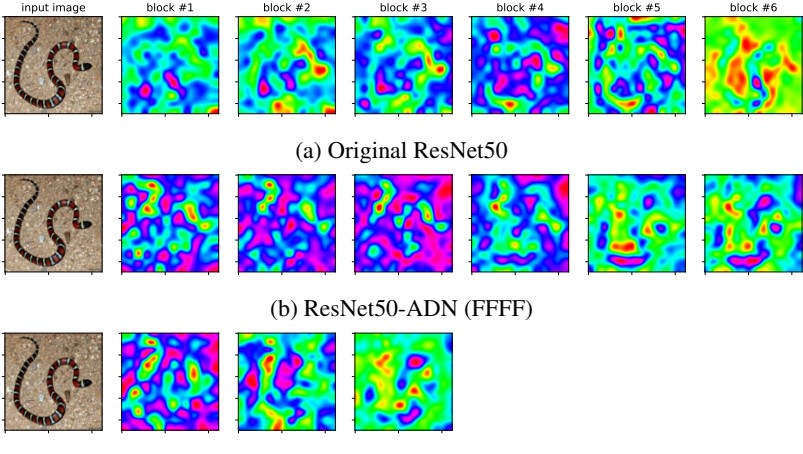

Figure 8: Class Activation Maps of the 3rd residual stages of ResNet50s. **(a)** Original ResNet50's activation regions change gradually across all blocks. **(b)** In ResNet50-ADN (FFFF), the first 3 blocks have extensive hot activation regions, implying active learning of new level features. In contrast, the skippable last 3 blocks have far less activation regions and they are gradually refined around the target. **(c)** Even though parameters are shared, the activation map of base-net is very different from super-net's since they use different batch normalization operators.

