# OpenReview forum: "Adaptive Depth Networks with Skippable Sub-Paths"
_NeurIPS.cc/2024/Conference — NeurIPS 2024 poster_

### Official Review · Reviewer_gteR · 2024-07-11

**Soundness:** 3
**Presentation:** 3
**Contribution:** 3
**Rating:** 7
**Confidence:** 4

**Summary:**

The authors propose an easy-to-train supernet, in which you can then adaptively change the network depth by disabling some blocks (skippable sub-paths) and using skip connections instead, while always keeping mandatory sub-paths. They found that the length of mandatory and the skippable sub-paths should be the same, for maximum efficiency in terms of accuracy and computational complexity.
During training there are 2 forward-backward passes:
1. Training the whole supernet/teacher (FFFF) network as usually for task objectives (ImageNet classificaiton)
2. Self-distillation intermidiate feature maps from supernet/teacher (FFFF) to the base/student (TTTT) network using KL-divergence-loss (where is base/student (TTTT) is a sub-network of supernet/teacher (FFFF) )

Switchable LayerNorm or BatchNorm operators are used in mandatory sub-paths.
This is the first approach to adaptive networks that provides a general principle and theoretic basis for supporting predictable depth adaptation with minimal performance degradation.

**Strengths:**

The resulting base/students (TTTT) network achieve higher accuracy than a separately trained networks with the same flops, depth and  structure.
The resulting supernet/teacher (FFFF) network achieves higher accuracy than a separately trained networks with the same flops, depth and  structure.
In Figure 4 (b), the proposed approach lies on the Pareto optimality curve, outperforming all other state-of-the-art approaches in terms of accuracy and computational complexity (Flops), except for SpViT-Swin-Ti, which is more accurate and has fewer flops.
The approach is very simple and in most cases it shows state of the art results. Controled experiments for different variants of the proposed approach are presented.
In Figure 5: (b) this is shown that the inference latecny of the proposed approach drops more than that of the S-ResNet50 approach with the same reduction in computational complexity.
It is tested on several widely used networks ResNet50, Swin-T and ViT-b/32, and show that it can be applied to both CNNs and transformers with minimal training effort.
Compared to methods of distillation, quantization, etc. this method is more flexible when, after training at test time, you can quickly increase the speed (at the expense of a decrease in accuracy) or increase the accuracy (at the expense of a decrease in speed) in a certain range, because at test time, sub-networks with various depths can be selected instantly from a single network.

**Weaknesses:**

In Figure 4 (b),  SpViT-Swin-Ti is more accurate and has fewer flops than the proposed approach Swin-T-ADN.

There is no Out-of-Domain (OoD) comparison when very different datasets are used for training and testing, which shows the robustness of the model and the extent to which it can be applied to real-world problems.
Most of the charts and tables compare the computational complexity (Flops), with the exception of Figure 5(b), while Latency (ms) is more important for real-world problems.
It would be better to have as many charts as possible with Out-of-Domain accuracy vs Latency.

**Questions:**

Does your approach achieve higher accuracy than  intermediate feature maps distillation: if you train the Supernet first, then freeze it and distill intermediate feature maps from the frozen Supernet to the separate randomly initialized Basenet? So maybe your approach is not only more flexible than intermediate feature maps distillation, but also more accurate?

**Limitations:**

Training and testing are carried out on different splits of the same dataset, which is an In-domain comparison, while for real-life applications Out-of-domain is much more important, when training and testing on very different datasets, where completely different approaches may be the best.

---

> ### Author Rebuttal · Authors · 2024-08-06
>
> * **Weaknesses a)**
> *In Figure 4 (b), SpViT-Swin-Ti is more accurate and has fewer flops than the proposed approach Swin-T-ADN.*
>
> Thank you for your positive feedback and valuable suggestion.
>
> While our work outperforms many state-of-the-art approaches, some efficient networks, notably SpViT-Swin-Ti, surpass some of our sub-networks on the Pareto front. SpViT-Swin-Ti excels because it better exploits transformer-specific features. For instance, SpViT-Swin-Ti dynamically prunes tokens, significantly reducing computation by eliminating less important tokens.
> In contrast, our depth adaptation approach is architecture-agnostic, making it applicable to both CNNs and transformers. We believe that our approach and SpViT-Swin-Ti are complementary. For example, SpViT-Swin-Ti could be trained to achieve better accuracy-efficiency trade-offs by skipping some encoder/decoder blocks, as suggested by our work.
>
> * **Weaknesses b)**
> *There is no Out-of-Domain (OoD) comparison ... It would be better to have as many charts as possible with Out-of-Domain accuracy vs Latency.*
>
> Out-of-domain (OoD) generalization in deep learning is crucial for real-world applications, as deployment data often differs from training data. However, achieving effective OoD generalization is challenging, and we are not aware of standardized benchmarks. We greatly appreciate the reviewer’s insights and would be grateful if references or benchmarks could be provided. This would significantly assist us in enhancing our approach for real-world applications.
>
> * **Question)**
> *Does your approach achieve higher accuracy than intermediate feature maps distillation: if you train the Supernet first, then freeze it and distill intermediate feature maps from the frozen Supernet to the separate randomly initialized Basenet? So maybe your approach is not only more flexible than intermediate feature maps distillation, but also more accurate?*
>
> The reviewer asked about the effectiveness of applying knowledge distillation (KD) from a super-net, such as ResNet50-ADN(FFFF), to a separate ResNet50-Base model.
>
> Firstly, our observations indicate that the naïve application of KD is not effective when using large datasets like ImageNet. In Figure 4(b), we illustrate the impact of applying KD to equivalent networks. Contrary to common belief, the naïve application of KD does not enhance performance. In fact, following the same training schedule (150 epochs for ResNets), KD results in worse performance compared to ordinary training using target labels. For instance, ResNet50-Base (KD individual), which is trained using pytorch pretrained ResNet50 as a teacher, achieves only 73.8% Acc@1, which is 1.2% lower than ResNet50-Base trained without KD. This finding aligns with prior work, such as [52][53][54], which also indicates that achieving positive results with KD on ImageNet is very challenging. To obtain positive outcomes with KD, an extended training schedule and a right combination of teacher/student and optimization techniques are required.
>
> Next, we conducted an additional experiment to investigate the effectiveness of exploiting ResNet50-ADN(FFFF), which was trained using our self-distillation strategy, as a teacher to train a separate ResNet50-Base model. The results show that ResNet50-Base trained using ResNet50-ADN(FFFF) as a teacher achieves 76.30% Acc@1, which is 0.2% higher than our subnetwork ResNet50-ADN (TTTT).
> This result demonstrates that the right combination of teacher and student is crucial for effective knowledge distillation. We conjecture that since our self-distillation process enforces ResNet50-ADN (FFFF) to produce intermediate features (and logits) compatible with ResNet50-ADN (TTTT), the knowledge was more effectively transferred to ResNet50-Base, which has the same architecture to ResNet50-ADN(TTTT). This is an interesting result and requires further investigation.
>
> The following table summarizes the evaluation results.
> | Teacher | Student | Student Acc@1 |
> |---|---|---:|
> |ResNet50 (Pytorch pretrained) | ResNet50-Base |  73.8% |
> |ResNet50-ADN(FFFF) | ResNet50-Base |  76.3% |

---

> > ### Comment · Reviewer_gteR · 2024-08-12
> >
> > Thanks to the authors for the answers and clarifications.
> > Based on the answers, I leave the article rating 7: Accept.
> >
> > "The results show that ResNet50-Base trained using ResNet50-ADN(FFFF) as a teacher achieves 76.30% Acc@1, which is 0.2% higher than our subnetwork ResNet50-ADN (TTTT). "
> > Did you use only output logits/pseudo-labels or also feature map tensors from **intermediate layers** (for example, after each downsampling layer) for distillation in this case?

---

> > > ### Author Response · Authors · 2024-08-13
> > >
> > > We sincerely appreciate your positive feedback and questions.
> > >
> > > In this experiment, we explored two scenarios for knowledge distillation (KD) of ResNet50-Base: (1) using only logits, and (2) using both logits and intermediate features. As shown in the table below, both scenarios yield slightly better performance compared to our self-distillation method. While the difference between (1) and (2) is marginal, there may be a chance to enhance the performance of (2) with further hyper-parameter tuning, such as applying different KD temperatures for different layers. However, we have not yet explored this path.
> > >
> > > This finding highlights the importance of selecting the right teacher-student combination for effective knowledge distillation. For instance, KD using ResNet50 (PyTorch pretrained) as a teacher did not yield positive results.
> > >
> > > | Teacher                                 | Student          | Student Acc@1 | Note |
> > > |-----------------------------------------|------------------|---------------:|-------:|
> > > | ResNet50(FFFF)       | ResNet50(TTTT)   | 76.1          |  our approach |
> > > | ResNet50 (PyTorch pretrained), logits only | ResNet50-Base    | 73.8          | |
> > > | (1) ResNet50-ADN(FFFF), logits only     | ResNet50-Base    | **76.3**          | |
> > > | (2) ResNet50-ADN(FFFF), logits + intermediate features | ResNet50-Base    | 76.2          | |
> > >
> > > Thank you again for your comments, and please let us know if you have any further questions.

---

### Official Review · Reviewer_RAbE · 2024-07-12

**Soundness:** 3
**Presentation:** 3
**Contribution:** 3
**Rating:** 5
**Confidence:** 2

**Summary:**

The submission presents an approach to adaptive depth networks, where each hierarchical residual stage is divided into two sub-paths and they are trained to acquire different properties:  the first sub-path is essential for hierarchical feature learning, the second one is trained to refine the learned features and minimize performance degradation even if it is skipped. In addition, a formal reason of why the proposed training method can reduce overall prediction errors while minimizing the impact of skipping sub-paths is also provided. Experimental results on ImageNet classification are provided to prove the effectiveness.

**Strengths:**

There are 2 major innovations in this submission:
1. Training Sub-Paths with Self-Distillation
2. Skip-Aware Batch/Layer Normalization
In the ablation study, both of these two innovations show improvements.
The strength of this submission is that it does not only present the ideas but also analytically proved the effectiveness.

**Weaknesses:**

The overall contribution of the paper is solid. The only thing I would suggest is to expend the experiments to 1 more low-level vision task, for example Single-Image Super Resolution, image denoising and so on. Since usually the decoder part of such low-level vision task is computational heavy.

**Questions:**

I don't have specific questions to this submission.

**Limitations:**

No problem in limitations.

---

> ### Author Rebuttal · Authors · 2024-08-06
>
> **Weaknesses)** *The overall contribution of the paper is solid. The only thing I would suggest is to expend the experiments to 1 more low-level vision task, for example Single-Image Super Resolution, image denoising and so on. Since usually the decoder part of such low-level vision task is computational heavy.*
>
> Thank you for your positive feedback and valuable suggestion. We appreciate your recognition of the solid contribution of our paper.
> We understand the importance of demonstrating the effectiveness of our approach across various vision tasks. Although we are not very familiar with Single-Image Super Resolution and image denoising, we acknowledge that our approach can be highly effective for computationally intensive tasks.
>
> Given the short rebuttal period, it is challenging to include the experimental results at this stage. However, we will strive to incorporate experiments on these tasks in the final version of the paper. We believe this will provide a more comprehensive evaluation of our method’s performance and computational efficiency.
>
> Thank you once again for your insightful feedback. We look forward to enhancing the overall impact of our paper with these additional experiments.

---

> > ### Comment · Reviewer_RAbE · 2024-08-13
> >
> > Fair. I don't require you to add new comparisons in this short rebuttal period. I tend to accept the submission even without it.

---

> > > ### Author Response · Authors · 2024-08-13
> > >
> > > Thank you once again for your valuable suggestions and understanding.

---

### Official Review · Reviewer_phTU · 2024-07-16

**Soundness:** 3
**Presentation:** 3
**Contribution:** 3
**Rating:** 5
**Confidence:** 3

**Summary:**

The paper presents adaptive depth networks. During training, the network is trained like a “super-net” that contains all the paths. During inference, the skippable networks can be skipped in devices that have limited resources. The whole framework is useful in real-world applications that require various accuracy-efficiency trade-offs.

**Strengths:**

- The motivation is clear. Proposing such a framework is useful in real-world applications.
- The authors conduct extensive experiments, including CNNs and Transformers.
- The experiments show that the proposed method is effective. Even on the TTTT setting, the performance drop is acceptable.

**Weaknesses:**

- There seem to be several solutions with ideas similar to those in this paper, such as [13]. Although there may be differences in the approaches, all of these methods can achieve the same goals in this paper. In my opinion, it would be better to include a comprehensive analysis of the differences and performance. If the proposed method could stand out in performance among the methods, it would make the proposed method very competitive among these methods.

- A recent study [R1] found that with longer schedules, the ResNet-50 model can achieve about 80% accuracy on ImageNet. Considering that we can choose to give smaller models a longer training schedule to achieve better results, is there a chance to use a longer schedule to make the Base model achieve better results and avoid using an adaptive network?


[R1] ResNet strikes back: An improved training procedure in timm.

**Questions:**

Please refer to the second point in weakness.

---

> ### Author Rebuttal · Authors · 2024-08-06
>
> * **Weakness a)** *There seem to be several solutions with ideas similar to those in this paper, such as [13]. ... it would be better to include a comprehensive analysis of the differences and performance. If the proposed method could stand out in performance among the methods, it would make the proposed method very competitive among these methods.*
>
> Thank you for your insightful and encouraging suggestion. We focused our approach on vision tasks due to the simplicity of the model architectures and training steps, which effectively highlight the advantages of our proposed method. However, as you rightly pointed out, the field of NLP has seen significant advancements recently, and there are some notable works such as [13] that shares the same goal. For broader impact, we need to compare our work to those works in NLP.
> Our method, which is applicable to transformer-based models in vision tasks, can indeed be extended to NLP models such as BERT and GPT. However, NLP models typically require more sophisticated training steps, including pretraining and fine-tuning, as well as substantial computational resources. We plan to explore these applications in a follow-up paper, where we will also investigate transformer-specific adaptation techniques.
>
> * **Question)** *A recent study [R1] found that with longer schedules, the ResNet-50 model can achieve about 80% accuracy on ImageNet. Considering that we can choose to give smaller models a longer training schedule to achieve better results, is there a chance to use a longer schedule to make the Base model achieve better results and avoid using an adaptive network?*
>
> As noted in [R1], the performance of ResNet can be enhanced by employing a longer training schedule and recent augmentation techniques, such as CutMix. Our ResNet50-ADN model benefits from these recent training techniques as well.
>
> Following table shows the results when we applied the PyTorch v2 training script (https://github.com/pytorch/vision/issues/3995)  to train ResNet50-ADN and individual networks. Using the PyTorch v2 training script, ResNet50-ADN (FFFF) and ResNet50-ADN (TTTT) achieve 80.44% and 78.78% top-1 accuracy on ImageNet-1K, respectively. The equivalent individual networks, ResNet50 and ResNet50-Base, achieve 80.44% and 78.17% top-1 accuracy, respectively.
> While ResNet50 (FFFF) and ResNet50 perform equally well, our smallest sub-network ResNet50 (TTTT) outperforms the equivalent ResNet50-Base by 0.62%. This is the result without active hyperparameter tuning, indicating that there is a chance for further performance improvement for our models.
>
> These results demonstrate that our approach can be effectively combined with recent training techniques.
> |   |   Acc@1 |
> |---|---|
> |ResNet50-ADN (FFFF) | 80.44%|
> |ResNet50-ADN (TTTT) | 78.79%|
> ||
> |ResNet50 (individual) | 80.44%|
> |ResNet50-Base (individual) | 78.17%|
>
> [R1] ResNet strikes back: An improved training procedure in timm.

---

### Official Review · Reviewer_dirV · 2024-07-22

**Soundness:** 3
**Presentation:** 4
**Contribution:** 3
**Rating:** 5
**Confidence:** 4

**Summary:**

The paper provides a training methodology to develop small deployable subnetworks (also high performing) when training a single large network. The key claim of this paper is that due to their innovative training techniques the smaller subnetworks learns better feature and proves the point with reasonable baselines.

**Strengths:**

Some of the key strengths of this paper is written below.

a) The results provided in this paper looks strong with a lot of baselines.

b) The depth adaptation analysis in sec 3 looks a bit hand wavy but tries to provide some prospective, which in general is a good practice.

c) The writing and presentation of this paper is very clear and concise.

**Weaknesses:**

Some major weaknesses of this paper are below.

a) The claim made in the paper " To the authro's knowledge, this is the first approach ..." is a bit of an overclaim. In prior CNN literature there are some works that performs subnetwork distillation.

b) In sec 3.2 the authors claim that h_{base} learns compact representation. I would request the authors to quantity this more concretely.

c) The results in Figure 4 a) are a bit tricky to evaluate as author's models are trained with distillation and the baseline models are trained with regular training. How do we know if the high accuracy is due to distillation or due to the proposed technique. The baseline has to be fixed.
d) More baseline comparisons are required with other techniques such as Matformer (https://arxiv.org/abs/2310.07707), Inheritune (https://arxiv.org/abs/2404.08634) etc.

**Questions:**

Can this method work in medium size LLMs where distillation could be tricky?

**Limitations:**

Yes, the authors have tried to address the limitations.

---

> ### Author Rebuttal · Authors · 2024-08-06
>
> * **Weakness a)** *The claim ... is a bit of an overclaim.*
>
> Thank you for your positive feedback and valuable suggestion.
>
> We appreciate the opportunity to clarify our claims. We acknowledge that there have been prior works in the CNN literature that perform sub-network distillation. Our intention was to highlight the effectiveness of our self-distillation strategy that focuses on training sub-paths, not sub-networks.
> To address your concern, we will revise the statement to more accurately reflect the novelty of our work in the context of prior studies.
>
> Revised statement: *“To the best of our knowledge, while prior CNN literature has explored sub-network distillation for adaptive networks, this approach uniquely provides a principle for training sub-paths for predictable depth adaptation. This principle allows us to avoid typical exhaustive training of target sub-networks and instead instantly construct sub-networks of varying depths from specifically trained sub-paths.”*
>
> We hope this revision clarifies our contribution and addresses your concern.
>
> * **Weakness b)** *Authors claim...$h_{base}$ learns compact representation. I would request the authors to quantity this more concretely.*
>
> We mention the *compactness* of $h_{\text{base}}$ because sub-networks, such as ResNet50-ADN(TTTT), achieve higher classification performance (e.g., by 1.1%) than equivalent ResNet50-Base, as shown in Figure 4-(a). We hypothesize that this performance improvement results from compressing the knowledge from $h_{\text{super}}$ to $h_{\text{base}}$ through the self-distillation process.
>
> Quantifying the compactness of representation is challenging, but we might attempt it by measuring the *cosine similarity* between feature representations $h_{\text{super}}$ and $h_{\text{base}}$. We can infer that the greater the similarity, the more knowledge has been transferred from $h_{\text{super}}$ to $h_{\text{base}}$. The following table shows the cosine similarity between $h_{\text{base}}$ and $h_{\text{super}}$ measured during forward passes of 1000 ImageNet validation data. At every residual stage, our approach demonstrates much higher similarity between $h_{\text{base}}$ and $h_{\text{super}}$, implying that more knowledge has been transferred to $h_{\text{base}}$.
>
> |  | Stage 1 |  Stage 2| Stage 3  | Stage 4 |
> |---|---:|---:|---:|---:|
> | ResNet50-ADN (ours) | 0.98 | 0.96| 0.81 | 0.87|
> | ResNet50 (Pytorch pretrained) | 0.91 | 0.82| 0.67 | 0.81|
>
> Additionally, in Figure 8 (Appendix B.2), we visualize $h_{\text{base}}$ and $h_{\text{super}}$ using Grad-CAM.
> These visualized images also show high similarity between $h_{\text{base}}$ and $h_{\text{super}}$.
>
> * **Weakness c)** *...How do we know if the high accuracy is due to distillation or due to the proposed technique....*
>
> In Figure 4(b), we illustrate the impact of applying knowledge distillation (KD) to equivalent networks. For instance, ResNet50 (KD individual) represents the performance when equivalent individual networks are trained with KD, using a PyTorch pretrained ResNet50 as the teacher network. Contrary to common belief, the naïve application of KD does not enhance performance. In fact, following the same training schedule (150 epochs for ResNets), KD results in worse performance compared to ordinary training using target labels.
>
> This finding aligns with prior work, such as [52][53][54], which also indicates that achieving positive results with KD on ImageNet is very challenging. To obtain positive outcomes with KD, an extended training schedule, a right combination of teacher/student and optimization techniques are required. For example, [52] achieves state-of-the-art ResNet50 performance with KD on ImageNet by employing a 1200-epoch training schedule.
> In contrast, our ResNet50-ADN trained with the proposed self-distillation strategy consistently achieves better performance than counterpart ResNets. This demonstrates that the high performance of adaptive depth networks does not simply come from distillation effect.
> |  |  Acc@1 |
> |---|---:|
> |ResNet50-ADN (FFFF) | 77.6 |
> |ResNet50-ADN (TTTT) | 76.1 |
> | | |
> |ResNet50 (individual) | 76.7 |
> |ResNet50-Base (individual) | 75.0 |
> | | |
> |ResNet50 (KD individual) | 75.1 |
> |ResNet50-Base (KD individual) | 73.8 |
>
> * **Weakness d)** *More baseline comparisons are required ... such as Matformer, Inheritune*
>
> Thank you for suggesting related works. We reviewed both papers and found that Matformer shares a similar goal with our approach. Both Matformer and our method train a universal model from which smaller sub-networks can be extracted without additional training. We believe that Matformer’s nested FFN architecture could complement our work by providing more fine-grained adaptation of FLOPs and parameters within transformer blocks.
>
> However, the Matformer paper does not include detailed performance for sub-networks, such as FLOPs versus accuracy, and the source code is not available to reproduce the results. Therefore, it is challenging to include Matformer as a baseline.
> Consequently, we will compare Matformer in the Related Work section.
>
> * **Question)** *Can this method work in medium size LLMs where distillation could be tricky?*
>
> We have not applied our approach to large language models (LLMs) as we currently lack the resources to train them. However, our results demonstrate that our depth adaptation approach is effective for Vision Transformers (ViTs), which utilize transformer encoders. Although not included in the paper, we recently applied our approach to DETR [R1], an object detection network with transformer decoders, and achieved depth adaptation without any loss of mean Average Precision (mAP).
> While we have not yet had the opportunity to apply our method directly to LLMs, we believe it can be effective since LLMs also consist of transformer encoders/decoders.
>
> [R1] Carion et al., “End-to-End Object Detection with Transformers”, ECCV 2020.

---

> ### Comment · Reviewer_dirV · 2024-08-12
> **Thank you Authors for the Rebuttal**
>
> a) The authors' has just updated the statement from subnetwork to subpath. But haven't differentiated it why these two are conceptually different. I have highlighted cause "later layers distilling knowledge to initial layers or a subset of  layers" is not novel. Please clarify how these are conceptually different.
>
> b) If the compactness of representation is difficult to prove then authors should refrain from making such claims. I could not understand the table; a self contained caption is needed. What are the values? what are the stages? how are they trained? how the evaluation is done? A layerwise confusion matrix would have been a better substitute to establish this correlation.
>
> c) I do not agree; KD has shown great results with Imagenet-1K and Imagenet-21K for instance look at DEIT [1] and TinyVIT [2]. I guess the author's are comparing with their version of KD. I request the authors to find the SOTA KD for such a setting. I acknowledge the table shown by the authors but I would request the authors to cite papers for their KD baselines.
>
> d) I acknowledge the Matformer discussion but the authors didn't say anything about the Inheritune paper.
>
> Question- Fair!! such an experiment would take time and rebuttal period is too short.
>
> Overall, this is a good work but some of my concerns that I have raised remain unanswered/ambiguously answered. I would retain my score. Again I re-iterate this is a good work but loose ends needs to be tightened with reasonable baselines and claims.
>
> [1] https://arxiv.org/abs/2012.12877
>
> [2] https://arxiv.org/abs/2207.10666

---

> ### Author Response · Authors · 2024-08-13
>
> * **Comments a) and b)**
>
> We appreciate your insight. After considering your feedback carefully, we realized that some of our initial statements might be misleading. Specifically, our initial claim that ‘$h_{base}$ learns a compact representation from $h_{super}$’ can be misleading. Since, in our approach, every sub-network shares parameters and $h_{super}$ is not fixed, the knowledge transfer is not uni-directional from $h_{super}$ to $h_{base}$, but rather bidirectional. As demonstrated in Section 3.3, the goal of our self-distillation strategy is to encourage $h_{base}$ and $h_{super}$ to learn similar feature distributions, rather than unidirectional knowledge distillation. Since $h_{base}$ and $h_{super}$ have similar distributions, the selected layers of every residual stage become skippable with minimal performance loss.
>
> In the revision, we will remove the claim that is hard to quantify and potentially misleading.
> In light of this, we will also rephrase our contribution as follows:
>
> *Our approach uniquely introduces a principle for training selected sub-paths to be skippable with minimal performance loss. This principle allows us to avoid typical exhaustive training of target sub-networks and instead instantly construct sub-networks of varying depths from specifically trained sub-paths.*
>
> * **Comment c)**  *I do not agree; KD has shown great results with Imagenet-1k and ImageNet-21K...*
>
> We appreciate your feedback and agree that knowledge distillation (KD) is a crucial technique for developing efficient networks. Previous works, such as DEIT [1] and TinyVIT [2], have demonstrated its effectiveness.
> In Figure 4-b, we compare our work with KD to illustrate that our positive results are not solely due to distillation effect. To ensure a fair comparison, we tried to maintain identical training settings for both our approach and KD. Our objective was not to claim that our method is superior to KD in general.
>
> In the revision, we will include efficient KD approaches, such as DEIT [1] and TinyVIT [2], in Table 1 as state-of-the-art (SOTA) baselines.
>
> * **Comment d)** *... but the authors didn't say anything about the Inheritune paper.*
>
> Inheritune shares some similarities with our approach as it leverages a few transformer blocks from a larger language model (LM) to train smaller models. However, unlike our method, Inheritune trains separate smaller models using the larger LM, making it not directly comparable to our approach. Additionally, a quantitative comparison between Inheritune and our method is not feasible since they are applied to different domains. We may consider Inheritune as a baseline if our work is extended to large language models (LLMs).
>
> * **Final Comment)**  *Again I re-iterate this is a good work but loose ends needs to be tightened with reasonable baselines and claims.*
>
> Thank you once again for your valuable suggestions and comments.
>
> They have given us an excellent opportunity to reflect on our work, considering both its strengths and weaknesses. We will update our revision based on your feedback.

---

### Author Rebuttal · Authors · 2024-08-06

Dear reviewers,

We sincerely thank all reviewers for their positive feedback and constructive suggestions.
We have made every effort to address each question and suggestion in detail.
Specifically, we conducted three additional experiments to respond to the reviewers’ questions and suggestions.

1.	**Quantifying compactness of $h_{base}$**: To quantify the compactness of $h_{base}$, we measured *cosine similarity* between $h_{base}$ and $h_{super}$ at every stage. The result shows that our models manifest higher similarity between two feature representations. Through this, we can infer that more knowledge was transferred from $h_{super}$ to $h_{base}$ with our self-distillation strategy.
2.	**Effect of a longer training schedulep**: One reviewer asked whether training a smaller network, such as ResNet50-Base, with a long training schedule [R1] could achieve better results and avoid the need for an adaptive network. To address this question, we applied the PyTorch v2 training script (https://github.com/pytorch/vision/issues/3995) to train ResNet50-ADN and individual networks. The results show that ResNet50-ADN(FFFF) and ResNet50-ADN(TTTT) achieve 80.44% and 78.78% top-1 accuracy on ImageNet-1K, respectively. The equivalent individual networks, ResNet50 and ResNet50-Base, achieve 80.44% and 78.17% top-1 accuracy, respectively. While ResNet50 (FFFF) and ResNet50 perform equally well, our smallest sub-network ResNet50 (TTTT) outperforms the equivalent ResNet50-Base by 0.62%. These results demonstrate that our ResNet50-ADN model benefits from these recent training recipes as well.
3.	**Effectiveness of ResNet50-ADN(FFFF) as a teacher**: We investigated the effectiveness of using ResNet50-ADN(FFFF) as a teacher to train a separate ResNet50-Base model. The results show that ResNet50-Base trained with ResNet50-ADN(FFFF) as a teacher achieves higher accuracy than our subnetwork ResNet50-ADN(TTTT). In contrast, applying knowledge distillation (KD) to train ResNet50-Base using a vanilla ResNet50 as a teacher does not yield positive outcomes. We conjecture that since our self-distillation process enforces ResNet50-ADN(FFFF) to produce features compatible with ResNet50-ADN(TTTT), the knowledge was more effectively transferred to ResNet50-Base.


Despite those efforts, due to the short rebuttal period, some suggestions remain for follow-up work. In particular, we received valuable suggestions about applying our approach to different tasks and domains. We recognize their importance for broader impact and will try to accommodate these suggestions in the final version of the paper and in follow-up works.
We deeply appreciate the reviewers’ efforts and insights.

Best regards,

[R1] Wirghtma, et al, “ResNet strikes back”, 2021

---

### Decision · Program_Chairs · 2024-09-25

**Decision:**

Accept (poster)

**Comment:**

The paper presents valuable contributions to the design of adaptive depth networks and tests the approach on both convolutional and transformer networks. After rebuttal, all the reviewers were positive about the paper, and the AC agreed with the rating.